# THE LIMITS OF INFERENCE SCALING THROUGH RESAMPLING

**Benedikt Stroebl   Sayash Kapoor   Arvind Narayanan**
Princeton University
`{stroebl,sayashk,arvindn}@princeton.edu`

## ABSTRACT

Recent research has generated hope that inference scaling, such as resampling solutions until they pass verifiers like unit tests, could allow weaker models to match stronger ones. Beyond inference, this approach also enables training reasoning models, where data is curated using rejection sampling against a verifier. However, we show that this approach is fundamentally limited when verifiers are *imperfect* and have a non-zero probability of producing false positives. Resampling cannot decrease this probability, so it imposes an upper bound to the accuracy of resampling-based inference scaling, *regardless of compute budget*. Our analysis shows that there is a strong correlation between the model's single-sample accuracy and its false positive rate on HumanEval and MBPP, whose unit tests have limited coverage. Therefore, no amount of inference scaling of weaker models can enable them to match the single-sample accuracy of a sufficiently strong model. Empirical results show that optimal sampling attempts are often fewer than 10, as the negative utility of false positives outweighs benefits, bending inference scaling curves downward. Finally, false positives may have other undesirable qualities, like poor adherence to coding style conventions.

## 1 INTRODUCTION

Scaling the amount of compute used during inference is a promising way to improve LLM performance. Techniques include reasoning (Wei et al., 2023; Wu et al., 2024; Setlur et al., 2024), reflecting on model outputs to revise candidate solutions (Shinn et al., 2023; Zhong et al., 2024), and compositions of these and other atomic techniques (Saad-Falcon et al., 2024; Welleck et al., 2024).

Inference scaling through *resampling* stands out for its simplicity and broad applicability. It works by generating many candidate outputs until one is satisfactory, based on feedback from a *verifier* (Song et al., 2024; Qin et al., 2024; Brown et al., 2024; Hassid et al., 2024; Li et al., 2024a). Unlike techniques such as majority voting where gains from inference scaling quickly plateau (Table 1), resampling has given rise to the hope of usefully scaling inference compute by many orders of magnitude.

We provide evidence that tempers this assumption. Our key concern is that the generalization gap — where a model performs well on benchmarks but fails to generalize to the real-world — is amplified when using repeated sampling to lift the performance of weaker models.

Specifically, we study the use of unit tests as verifiers for coding benchmarks, to see if inference scaling for less capable models allows us to match the accuracy of more capable models. We make the following contributions.

**Review of inference scaling techniques and their limitations (Section 2).** We review papers on inference scaling, categorizing the primary techniques and listing their domain-specific applications and known limitations.

**Demonstration of generalization gap (Section 3).** We provide empirical evidence on two benchmarks, HumanEval+ and MBPP+ (Liu et al., 2023b), showing that the apparent gains from resampling with imperfect verifiers are unlikely to translate into real-world performance. Despite achieving comparable results to stronger models on standard unit tests, less capable models suffer from a larger

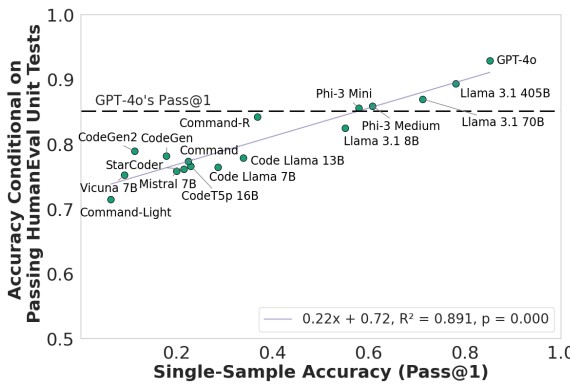 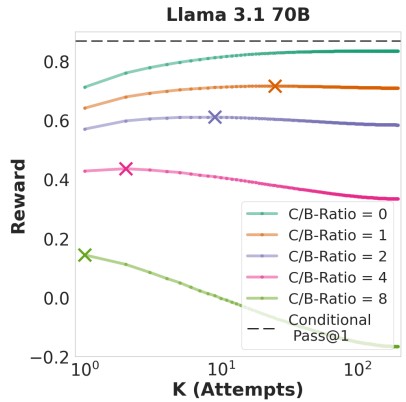

**Single-sample accuracy vs. resampling limits on HumanEval+.** The x-axis shows single-sample accuracy on HumanEval+ (which contains comprehensive unit tests), while the y-axis shows the highest achievable accuracy when resampling with an infinite compute budget, using HumanEval's more limited unit tests as verifiers. Weaker models (models with lower single-sample accuracy) produce false positive solutions at higher rates. Models below the cutoff line are unable to match GPT-4o through resampling, as GPT-4o's Pass@1 exceeds the accuracy of such a model even when conditioned on its solutions passing the unit tests. Results on MBPP+ follow a similar pattern (Figure 3).

**The cost of false positives limits the reward of resampling.** False positives w.r.t. an imperfect verifier (HumanEval unit tests) incur a "cost" (e.g., subtle bugs in code), while correct answers provide a benefit. The reward (y-axis) depends on this cost-benefit ratio (C/B-Ratio). Problem instances that require more attempts tend to be harder, hence more susceptible to false positives. Thus, even with *zero computational cost*, for realistic cost-benefit ratios, the optimal number of samples $K$ is finite and very low.

Figure 1: Overview of our main findings.

generalization gap—producing incorrect solutions that fail the extended test suite (false positives) at higher rates than stronger models.

In particular, we observe that even if given an *infinite* inference budget, in many cases a weaker model cannot match the performance of a single invocation of a sufficiently strong model.

**Empirical analysis to understand the limitations of inference scaling with imperfect verifiers (Section 4).** We examine how introducing a cost (negative utility) for returning false positives impacts the optimal number of resampling attempts on HumanEval+. We find that even with an infinite inference budget, the optimal number of samples is often finite and very low (e.g., $K \leq 5$ in Figure 4). Hence, resampling quickly reaches a point of diminishing returns without bridging the performance gap for smaller models. If the cost of an incorrect solution is higher than the benefit of a correct solution, the optimal $K$ can be *zero* — the risk of a false positive for a weak model is high enough that it is effectively useless (Figure 4). In Section C we present a theoretical model that complements the findings in this section.

**Evidence that this affects code quality beyond correctness (Section 5).** We show that the reliance on imperfect verifiers not only affects the functional correctness but also overall quality of the generated code. We evaluate candidate solutions on HumanEval+ based on various readability metrics such as adherence to naming conventions (that we specify in the prompt) like `snake_case` and `camelCase`, line-level commenting, and guidelines regarding the maximum line length and number of lines in function implementations. We find that false positive solutions are lower quality across all models and metrics when compared to true positive solutions. While other aspects of code quality such as simplicity and modularity are harder to test automatically, we speculate that the same pattern holds for those properties as well.

We also conduct a qualitative analysis to identify recurring error types causing a larger generalization gap for weaker models.

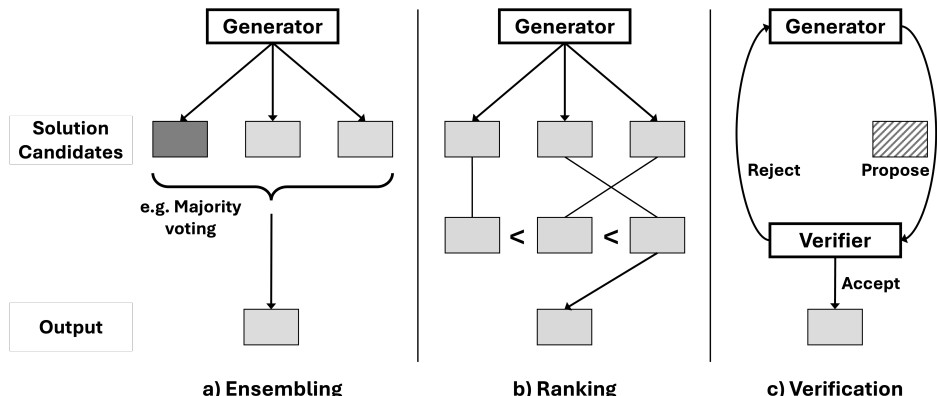

Figure 2: Schematic illustration of various resampling techniques for inference scaling.

Our findings have three additional implications. First, they show the importance of building highly accurate verifiers. This goal might benefit from treating verification technology as a specialized subfield with its own metrics and benchmarks. This is especially true for training-time uses: models trained with feedback from imperfect verifiers may learn to exploit weaknesses in the verifier rather than solve tasks robustly, potentially leading to safety concerns (Krakovna et al., 2020; Amodei et al., 2016).

Second, the use of imperfect verifiers as the ground truth for evaluation is flawed. We used HumanEval+ and MBPP+ for evaluation; the gaps we identify would have been invisible if we had used HumanEval and MBPP both as verifiers and as benchmarks. While the limitations of these benchmarks for measuring absolute performance are well known (Zhang et al., 2024b; Liu et al., 2023b), our results show that they might result in misleading comparisons between models as well.

Third, our findings highlight limitations in resampling-based data curation for reasoning models. Reasoning models rely on datasets curated through rejection sampling against verifiers. When these verifiers are imperfect, the curated datasets risk including mislabeled examples, which incurs a cost on model performance. This introduces a bottleneck: without stronger base models or highly accurate verifiers, the gains from resampling-based data curation to train reasoning models are likely limited.

While we do not claim that resampling is the predominant scaling technique, our findings suggest that the persisting gap between oracle and imperfect verifiers should be taken seriously and could pose limitations across inference scaling strategies. We invite research on ways to mitigate the issues identified in this work.

## 2 SCALING INFERENCE COMPUTE WITH VERIFIERS

Table 1 provides an overview of the main techniques for scaling inference compute with LLMs. Some methods such as majority voting (Wang et al., 2023b; Chen et al., 2024a) or resampling using verifiers (Brown et al., 2024; Xin et al., 2024) generate many candidate solutions and then select one. Other methods such as reasoning (Wei et al., 2023) and critique (Shinn et al., 2023; Madaan et al., 2023) refine a single solution. In practice, these methods can be combined in flexible ways and the distinction between them is not always clear (Section A.1). Note that our notion of inference scaling excludes methods such as those used to train OpenAI's o1 series of models, since we are only looking at improvements during inference time to available language models, rather than training improvements.

All these methods except verifier-based resampling are known to have important limitations that cast doubt on how much scaling is truly possible, as summarized in Table 1. Resampling using verifiers has a different control flow than other methods, which gives it an intuitive appeal (Figure 2): we can potentially regenerate solutions *indefinitely* until one is correct. This enthusiasm around resampling is partly driven by the empirically observed *inference scaling laws*, which suggest that the fraction of tasks for which we find at least one correct solution scales predictably with the number of samples over multiple orders of magnitude (Brown et al., 2024).

Table 1: **Overview of inference scaling techniques.** This table shows the main categories of techniques for inference scaling along with their descriptions and known limitations. Note that rankers, often implemented with reward models, are sometimes referred to as verifiers and the boundary can be unclear.

| Technique | Description | Limitations |
|---|---|---|
| Ranking | Scores and ranks the best samples from multiple candidates (Cobbe et al., 2021; Hassid et al., 2024; Liu et al., 2023a; Lightman et al., 2023; Liu et al., 2023c; Hosseini et al., 2024b; Kirchner et al., 2024; Setlur et al., 2024; Snell et al., 2024; Vacareanu et al., 2024; Chen et al., 2024c) | • Doesn't scale with sample budget (Brown et al., 2024)
• Underperform compared to other methods (Zhang et al., 2024a) |
| Majority Voting | Using consensus among multiple samples to determine final output(Wang et al., 2023b; Li et al., 2024b; Wang et al., 2024b) | • Hurts performance on hard tasks and non-monotonous scaling under task heterogeneity (Chen et al., 2024a)
• Sample inefficient for queries with many answer possibilities (Wang et al., 2024b)
• Limited applicability for tasks with non-discrete answers |
| Oracle Verification | Leverages ground-truth evaluator for free until correct solution is found (Xin et al., 2024; First et al., 2023) | • Not available for most domains |
| Imperfect Verification **(This paper)** | Scores and accepts or rejects candidate solutions (Zhang et al., 2024a; Davis et al., 2024; Yao et al., 2023; Gundawar et al., 2024; Kambhampati et al., 2024) | • Bigger generalization gap for weaker models (Section 3)
• Optimal number of samples is finite and low (Section 4)
• Low code quality of false positives (Section 5) |

However, the usefulness of this depends on the availability of a capable verifier (Davis et al., 2024). In some settings, we may have an *oracle verifier*, such as a proof checker, that does not suffer from false positives — that is, if the proof checker verifies the proof, it is guaranteed to be correct. False negatives of the verifier (including nontermination under a fixed compute budget) are less of a problem, as one can simply generate more samples until a true positive is found. It is possible that *every* correct solution is a false negative of the verifier, but it is unclear if this is a problem that arises in practice.

But in other settings such as coding and reasoning, we only have *imperfect verifiers* such as unit tests or LM judges, which suffer from false positives: incorrect solutions that nonetheless pass the verifier. In these settings, we don't have easy methods to guarantee the correctness of generated solutions at inference time. As a result, we cannot distinguish between false positives and true positives simply by increasing the compute budget. We survey papers that use verifiers in Table 2.

In this paper, we investigate the effect of scaling inference compute with access to imperfect verifiers. We distinguish ranking from verification: ranking selects the best among multiple candidates, while verification accepts or rejects each candidate independently against a correctness criterion.

## 3 REPEATED SAMPLING WITH WEAKER MODELS LEADS TO WORSE GENERALIZABILITY

In computer programming tasks, unit tests are commonly employed as verifiers to assess the correctness of candidate solutions generated by language models. While unit tests are practical and efficient, they often suffer from imperfect test coverage, leading to false positives where incorrect solutions pass the tests (Gulwani et al., 2017). This affects many benchmarks such as HumanEval (Chen et al.,

| Paper | Verifier Category | Verifier Type | Domain | Verifier Implementation |
|---|---|---|---|---|
| Chen et al. (2022) | Imperfect | Unit tests | Coding | Checks agreement of tests and samples |
| Shinn et al. (2023) | Imperfect | LM-as-judge, Unit tests | Coding, QA | LLM evaluator generates decision rewards |
| Yao et al. (2023) | Imperfect | LM-as-judge | Planning | LLM evaluates reasoning steps |
| First et al. (2023) | Oracle | Proof checker | Math | Proof checker |
| Thakur et al. (2024) | Oracle | Proof checker | Math | Proof checker |
| Yang et al. (2023) | Oracle | Proof checker | Math | Proof checker |
| Wang et al. (2023a) | Oracle | Proof checker | Math | Proof checker |
| Azerbayev et al. (2024) | Oracle | Proof checker | Math | Proof checker, Majority voting |
| Huang et al. (2024) | Oracle | Proof checker | Math | Proof checker |
| Xin et al. (2024) | Oracle | Proof checker | Math | Proof checker |
| Brown et al. (2024) | Oracle | Proof checker | Math | Proof checker |
| Davis et al. (2024) | Imperfect | LM-as-judge | QA, Math | LLM judges correctness of generations |
| Hassid et al. (2024) | Imperfect | Unit tests | Coding | Unit tests |
| Zhang et al. (2024a) | Imperfect | Generative RM | Math | Verification as part of the model output |
| Zhuge et al. (2024) | Imperfect | Agent-as-judge | Agents, Coding | Agents evaluate outputs of other agents |
| Kapoor et al. (2024) | Imperfect | Unit tests | Coding | Unit tests |
| Saad-Falcon et al. (2024) | Imperfect | LM-as-judge | Coding, Reasoning | LLM judges correctness of generations |
| Liang et al. (2024) | Imperfect | Program-of-thought | Coding, Math | Checks CoT against generated PoT |
| Cook et al. (2024) | Imperfect | LM-as-judge | Instruction-following | LLM checks answer against generated checklists |
| Gundawar et al. (2024) | Imperfect | Agent-as-judge | Travel planning | Pre-defined constraints verified by critic agents |

Table 2: Survey of papers on LLM verification methods, their approaches, and specific verifier implementations.

2021), APPS (Hendrycks et al., 2021), or MBPP (Austin et al., 2021). This imperfection raises the question: Do less capable models produce false positives—implementations that pass the standard unit tests but fail the comprehensive ones—at a higher rate than stronger models?

**Experimental setup.** To investigate this, we conducted experiments on two widely used coding benchmarks: HumanEval+ and MBPP+. MBPP consists of simple programming tasks designed to evaluate the basic coding abilities of models (Austin et al., 2021). HumanEval+ and MBPP+ are extensions of the original HumanEval and MBPP benchmarks (Liu et al., 2023b) and contain additional hidden test cases to assess correctness beyond the unit tests included in the original benchmarks.

We evaluated models of varying capabilities, including weak and stronger models, generating at least 200 samples per task and model on HumanEval+ (1,000 for Command Light, to minimize tasks without any passing solutions) and 50 samples per task on MBPP+. To assess the generalization gap, we then evaluated solutions that passed the original benchmark test sets on the more comprehensive hidden test cases (see Section D for details). These tests are extensive, and we assume that solutions that pass the full set of tests are correct. (If this assumption is not true, the generalization gaps that we reveal only grow bigger.)

**Findings.** Weaker models exhibit a higher probability of producing false positives compared to stronger models (Figure 3). This probability scales inversely with the true capability. This linear relationship holds with remarkable consistency across models of various families, including Cohere's Command models, GPT-4o, and the Llama-3.1 family. This suggests that while weaker models appear to perform well on standard benchmarks through increased sampling, they fail to generalize effectively and, importantly, they generalize worse than more capable models. They tend to generate fragile solutions that exploit the limitations of the unit tests. We speculate that this is because weaker models' "true understanding" of the programming tasks is worse.

The empirical results reinforce a core insight. Suppose $\mathbb{P}_{strong}(\text{Correct}) > \mathbb{P}_{weak}(\text{Correct}|\text{Pass Verifier})$. That is, the single-sample accuracy of a strong model exceeds that of a weaker model, even conditioned on the weaker model passing the base unit tests. Then the weaker

model cannot match the performance of a single invocation of the stronger model, no matter how big the compute budget for the weaker model. In Figure 3, this is shown by a horizontal line. No model below the line can match the performance of GPT-4o through resampling.

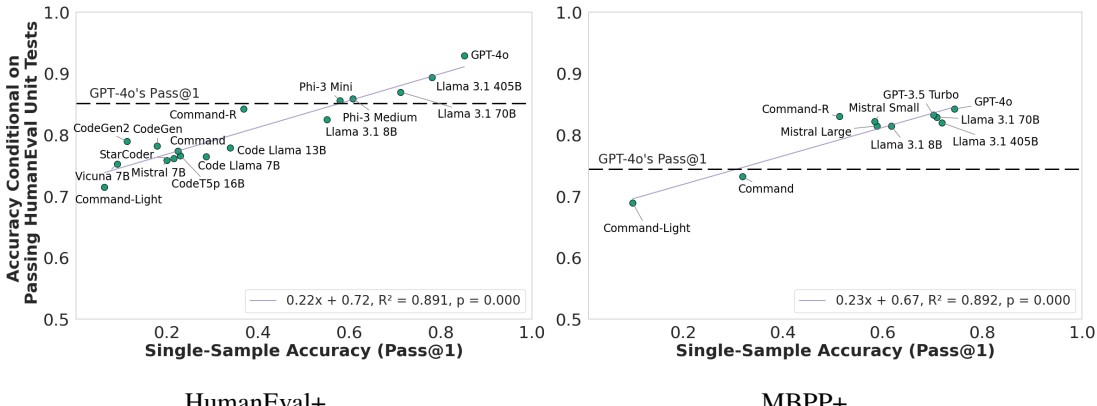

HumanEval+                                                 MBPP+

Figure 3: **Generalization gap with infinite compute budget.** We show the relationship between the accuracy of individual samples (x-axis) and the achievable accuracy given an infinite compute budget and limited unit tests (y-axis; note that it starts at 0.5). We evaluate performance on the extended test suites of HumanEval+ and MBPP+, using the unit tests from the original benchmarks as verifiers. For both benchmarks, the trend is that less capable models are more likely to generate false positives than stronger models. In Section B, we show our results with the full y-axis as well as upper (lower) bounds on the conditional accuracy accounting for tasks for which we did not observe *any* solutions passing the unit tests.

The effect is largely driven by a subset of the tasks where the unit tests are poor. When we limit the analysis to these tasks, the relationship is even more pronounced Section B.

Note that our results rely on human-generated unit tests as verifiers. In practice, we might expect to use language-model-generated unit tests for inference-time verification. It is an open question as to how the findings change when the unit-test verifiers are LLM-generated.

## 4   HOW MANY SAMPLES ARE OPTIMAL?

In the previous section we looked at the behavior of resampling in the limit as the number of samples grows large. Now we look at inference scaling curves, which allow us to study how accuracy varies as a function of the number of samples.

We add one important detail: we model the cost of false positives, such as code that passes unit tests but has subtle bugs. The cost of bugs (which might result in buggy software being deployed) is not easily comparable to the labor-saving benefit of correct solutions, and this cost-benefit ratio can vary greatly depending on the application. So we consider many possible values for the cost-benefit ratio, including zero, which is the setting considered in previous work on inference scaling. The ratio can potentially be much higher than 1 in some applications, such as security sensitive ones, since bugs might translate to exploitable vulnerabilities.

**Experimental setup.** For each model of interest, we generated 200 samples for each task in the HumanEval benchmark. For each K $\leq$ 200, If a passing solution was found within $K$ samples, we assigned rewards based on the outcome: a true positive yielded a benefit of 1, while a false positive incurred a cost, with values set according to different cost-benefit ratios: 0, 1, 2, 4, or 8 (Figure 4). If no passing solution was found within $K$ samples, we assigned a reward of 0 (both, cost and benefit are 0). We repeated this whole process 1,000 times and computed the mean reward for each $K$. The set of samples was the same in all 1,000 runs, but the order of samples was randomly permuted. This setup allows us to empirically observe the relationship between the number of sampling attempts $K$ and the reward for various cost-benefit ratios. The results are illustrated in Figure 4 for the Llama-3.1 (Dubey et al., 2024) and Code Llama (Rozière et al., 2024) model families. The scaling curves for GPT-4o are included in Figure 15.

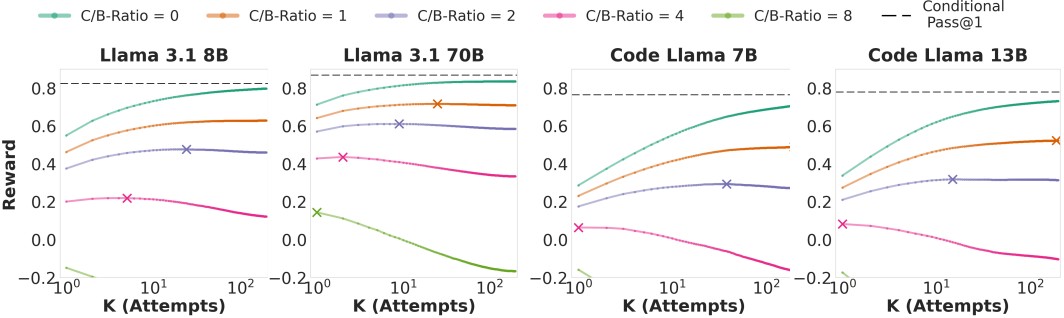

Figure 4: **Inference scaling curves in the presence of a cost for false positives.** We show the reward as a function of the number of attempts $K$ across various cost-benefit ratios for the Llama-3.1 and Code Llama model families. Crosses mark the optimal number of samples for each setting. Standard inference scaling curves with no cost (i.e., cost-benefit ratio is 0) are provided for reference. We find that, even at *zero computational cost*, there is a finite optimal number of samples $K$ that is often very low.

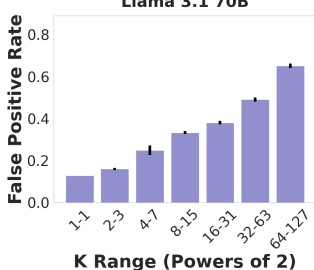
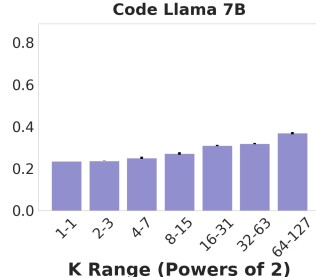

Figure 5: False positive rate as a function of the number of attempts $K$ for Llama 3.1 70B and Code Llama 7B on HumanEval+. We include plots for additional models in Section C.

**Findings.** The results show that the effectiveness of repeated sampling quickly reaches a point of diminishing and even negative returns. Each additional attempt brings a trade-off: although it might yield a correct solution, it might instead yield a false positive, and the false positive rate increases with $K$ (Figure 5). At first this is surprising, since sampling is a memoryless process. To understand why it happens, we need to look at the distribution of task difficulty (Figure 6), which turns out to be strongly bimodal. The easy tasks get solved within a few attempts, and for the remaining harder tasks, false positives are more likely. This aligns with findings by Chen et al. (2024a), who observed a similar inverse U-shaped accuracy curves explained by the heterogeneity in task difficulties.

Thus, even with *zero computational cost*, the optimal number of samples is finite and low (Figure 4). For example, at a cost-benefit ratio of 4, the optimal number of samples is $K \leq 5$ for all four models. If the ratio is high enough, the optimal number of samples is zero — the expected cost of a false positive outweighs the expected benefit of a correct solution, so the reward is always negative and it is best not to attempt a solution at all.

We note one important caveat: for some models such as Llama-3.1-70B, the false positive rate increases dramatically with $K$, whereas for others such as the Code Llama and Command families, the increase is much more gradual, resulting in much higher values of the optimal $K$, especially for low cost-benefit ratios. We have not been able to identify any intuitive reason for this difference.

To summarize, weaker models cannot "sample their way" to top-level performance if the verifier cannot reliably filter out false positives because the risks quickly outweigh the benefits. Our findings in this section align with our theoretical model in Section C, which generalizes these findings to other benchmarks.

## 5 FALSE POSITIVE SOLUTIONS ARE LOW-QUALITY EVEN BEYOND CORRECTNESS

While correctness is a fundamental criterion for evaluating code generated by LLMs, it is not the only determinant of code quality. High-quality code possesses attributes beyond mere functionality, such as readability, maintainability, and efficiency. Readability simplifies error-checking and is considered one of the most useful properties of high-quality code (Börstler et al., 2023). It can be measured using various metrics, including code length guidelines (e.g., PEP8), adherence to naming conventions like `snake_case` or `camelCase`, and consistent commenting (Zheng et al., 2024). Intuitively, shorter code with clear variable names and informative comments is generally easier to read and maintain.

To understand the relationship between imperfect verifiers and code quality, we evaluated the readability of code generated by various models in our setup.

**Experimental setup.** On HumanEval, we evaluate the readability scores of candidate solutions that pass standard unit tests and the more comprehensive test suite. For each measure of code readability, we use a different prompt instructing the model to adhere to the desired guidelines (see Section D for detailed prompts). We rely on the prompts and implementation from Zheng et al. (2024).

The results show notable differences in code quality between false positives and robust implementations. False positives, passing only the standard but not the extended unit tests, tend to have worse code quality across all metrics (see Figure 7). This trend is consistent across models of varying capabilities. This suggests that the limitations of imperfect verifiers for coding tasks extend beyond correctness issues but also affect other code characteristics important for software development. This affects weaker models more, given that they are more prone to generate false positives.

An open question arising from our findings is whether fine-tuning LLMs on code quality metrics could improve not only the quality of generated code but also robustness (Jain et al., 2023), potentially mitigating the prevalence of false positives.

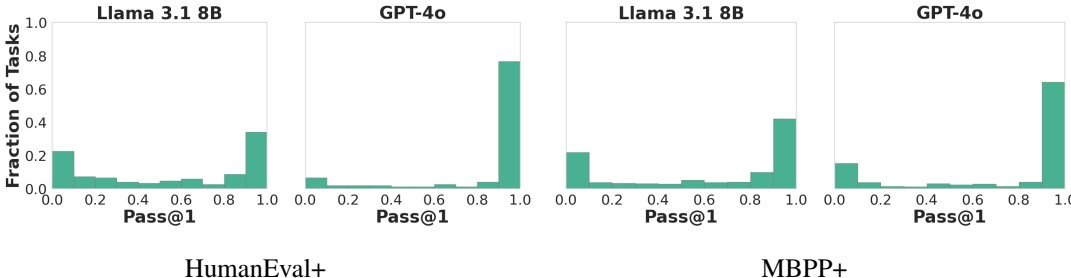

Figure 6: Distribution of task difficulties for Llama 3.1 8B and GPT-4o on HumanEval+ and MBPP+. We include barplots for all models on both benchmarks in Section B.

**Qualitative analysis of false positives.** To better understand the characteristics of false positive implementations, we randomly sample 10 implementations across all models that pass the standard tests but fail the additional unit tests, 5 for each benchmark. Through manual analysis, we identified several recurring error types. All examples mentioned in the following are included in Section D.3.

1. **Logical errors:** Such errors were common. For instance, in the HumanEval/30 task, the model is tasked with returning only positive numbers from a list. The solution shown in Section D.3 incorrectly converts floats to integers, passing the basic tests that included only integers but failing extended tests that introduce float values.

2. **Edge case handling:** Sometimes solutions failed to account for atypical inputs, which happened to not be covered by the standard unit tests. For example, in HumanEval/6, the solution failed to handle an empty list input. It is important to note that tasks in HumanEval and MBPP often are ambiguous as to how edge cases should be handled. For example, for HumanEval/149 some solutions fail because they return an assertion error instead of an empty list for the edge case of getting an empty list as input. We expect that ambiguity should affect weaker and stronger models similarly, but have not tested this.

3. **Inefficient implementations:** While most false positives result from logic errors or edge case mishandling, some were also caused by inefficient implementations. For instance, in HumanEval/15,

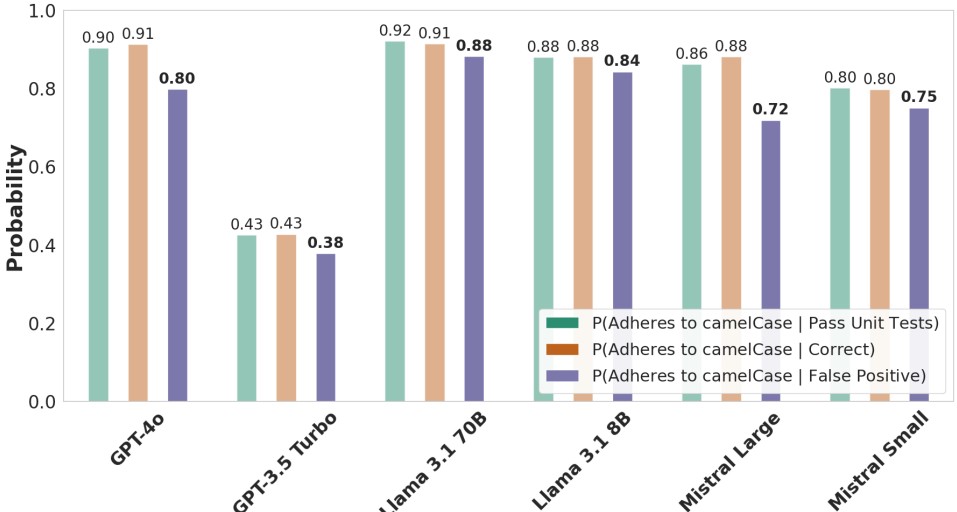

Figure 7: **False positives tend to be lower-quality code than correct implementations.** For example, false positive solutions fail to adhere to the `camelCase` naming convention more often than robust implementations. Figure 16 shows that this holds consistently across models and for all four code quality metrics we test. GPT-3.5 exhibits a low relative performance in using `camelCase` but performs comparably to other models in following `snake_case` (Section D). This has also been found in previous work (Zheng et al., 2024).

the solution involved a for-loop that became inefficient when handling larger inputs, causing a timeout on the extended tests. Following Liu et al. (2023b), we set the timeout such that each candidate solution must compute in less than one second or four times the time it takes to run each test on the ground truth implementation, whichever is greater.

## 6 DISCUSSION

We study a setting where all generators are paired with the same verifier. The verifier has imperfect coverage, but no false negatives. In real-world deployment settings, human-written unit tests are rarely available and we would need to rely on the use of automated test generation techniques. These approaches include symbolic execution (Lukasczyk & Fraser, 2022), specialized transformers (Tufano et al., 2021), and LLMs (Chen et al., 2024b; 2022; Siddiq et al., 2024). Model-generated tests introduce new challenges including a disparity between verifiers and a risk of false negatives. This could widen the generalization gap. We leave an investigation of the impact of model-generated unit tests as a next step.

Resampling is used not only to scale inference but also to train large reasoning models. Many state-of-the-art reasoning models are trained on datasets curated through rejection sampling (NovaSky Team, 2025; DeepSeek-AI et al., 2025; Bespoke Labs, 2025), where verifiers filter out incorrect outputs. However, imperfect verifiers can introduce mislabeled examples, implicitly incurring a cost of false positives. We hypothesize that weaker models paired with imperfect verifiers fail to produce datasets of sufficient quality to train competitive reasoning models, creating a bottleneck: without stronger base models or more accurate verifiers, gains from resampling-based data curation to train reasoning models are limited.

Although our experiments focus on coding tasks, we want to emphasize that our theoretical results are domain-agnostic: resampling relying on imperfect verifiers with non-zero false positive rates will face the same fundamental ceiling.

Related work has also studied the risks of over-optimizing against imperfect rewards. For example, Gao et al. (2022) analyzes how optimizing against proxy reward models can lead to degraded true

performance. Our setting differs in that we focus on inference-time resampling, where the ceiling on achievable accuracy arises directly from false positives rather than reward misalignment.

Finally, our findings weaken support for previous papers' claims that resampling is an effective strategy to increase accuracy by trading off inference time compute (Kapoor et al., 2024; Chen et al., 2024a); here resampling with imperfect verifiers is inherently limited.

**Limitations.** Our experiments focus on repeated sampling in the context of coding tasks. Coding offers a clear example of the challenges posed by imperfect verifiers, other domains might exhibit different behavior. Future work could extend these findings to tasks such as reasoning (Hosseini et al., 2024a), web agents (Bai et al., 2024; He et al., 2024), or agent-user interaction (Yao et al., 2024). Another limitation is prompt sensitivity, which affects LLM evaluations (Biderman et al., 2024; Liang et al., 2023). While we followed the original authors' implementation provided with the HumanEval+ and MBPP+ benchmarks (Liu et al., 2023b), prompt engineering could influence false positive generation. Additionally, we did not investigate how benchmark contamination affects our findings, as models could be overly optimized for passing the standard test cases. We did not explore mitigation strategies such refining solutions after they passed the verifier (Saad-Falcon et al., 2024). Similarly, we did not test alternative strategies to inference scaling that, e.g., induce more diversity during sampling (Wang et al., 2024a). Finally, the optimal number of samples K that we identify depends on the cost-benefit ratio, which varies by application context. While we show that K is finite and often very low across a range of ratios, we do not provide empirical evidence for what specific ratio applies in any given real-world deployment. In security-critical applications, the cost of a false positive may far exceed the benefit of a correct solution, whereas in lower-stakes settings a higher K may remain practical.

**Acknowledgments.** We thank Boris Hanin, Peter Henderson, Alexander Wettig, Jon Saad-Falcon, and Zachary Siegel for providing valuable feedback on earlier drafts of this work. We thank Cohere for their support of this work with API credits. We also thank Schmidt Sciences for their funding and support of this work.

**Reproducibility.** We release code to reproduce all experimental results of this paper in a GitHub repository[1]. This repository also contains all code samples for all models used in our experiments. Additionally, we provide an implementation of the theoretical model in Section C as a Python notebook.

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

# A    ADDITIONAL DETAILS ON SECTION 2

## A.1    EDGE CASES IN OUR GENERATOR-VERIFIER SETTING

The setting described in Figure 2 considers verifiers and generators as distinct components, where verifiers score and accept or reject individual samples from the generator's output to enable accuracy improvements through resampling.

This creates interesting edge cases with methods like Chain-of-Verification (CoVe) (Dhuliawala et al., 2023), Tree of Thoughts (ToT) (Yao et al., 2023), and Reward Models (RMs) (Snell et al., 2024) where verification and generation are more tightly coupled. While ToT fits within our framework by producing aggregate scores and a decision on whether the problem is solvable from a given state or not (i.e. potentially rejecting solutions), CoVe differs fundamentally in its verification approach. Instead of producing numeric scores and accepting or rejecting solution candidates, CoVe uses verification of intermediate facts used for answering a question to improve a single response through iterative refinement. This makes CoVe less suitable for inference scaling through resampling because there is no way to distinguish between the quality of multiple samples and using a verifier's verdict for resampling.

# B ADDITIONAL DETAILS ON SECTION 3

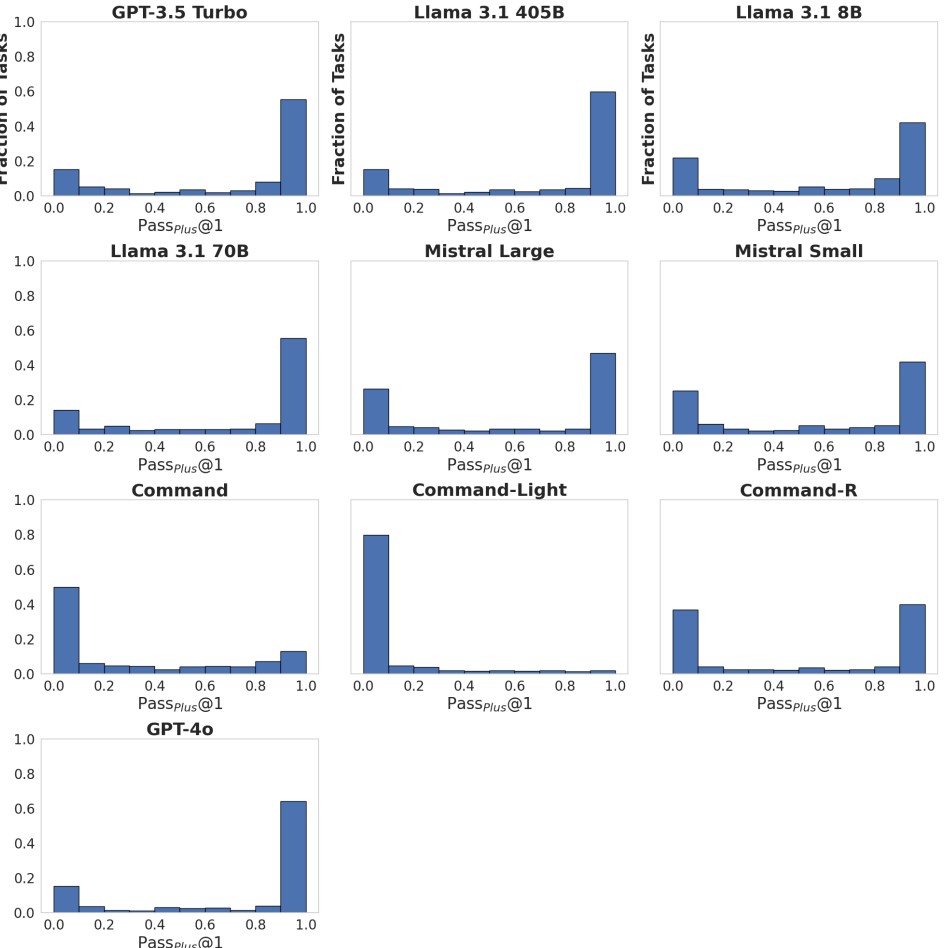

Figure 8: Distribution of task difficulties across models on MBPP+.

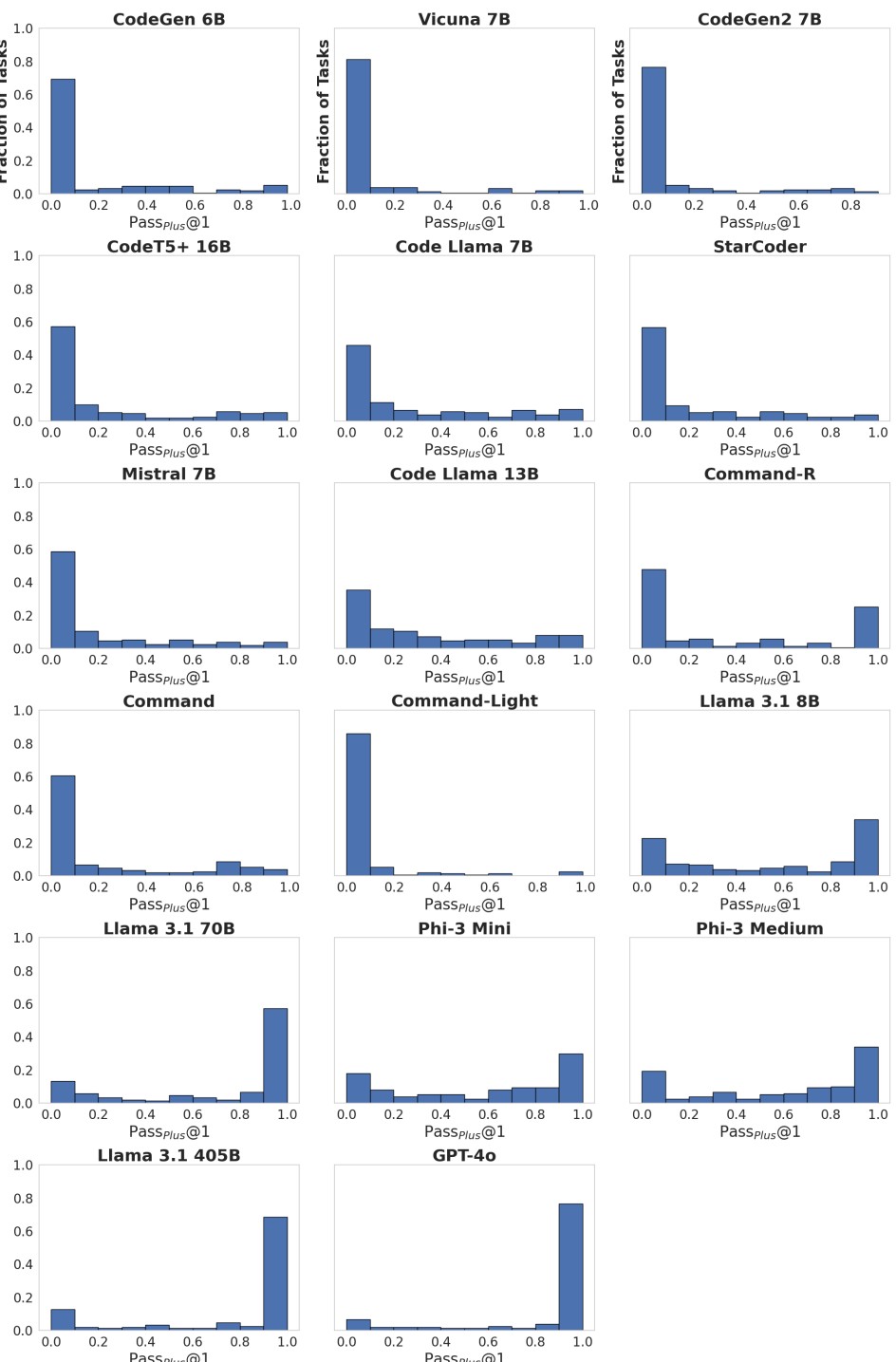

Figure 9: Distribution of task difficulties across models on HumanEval+.

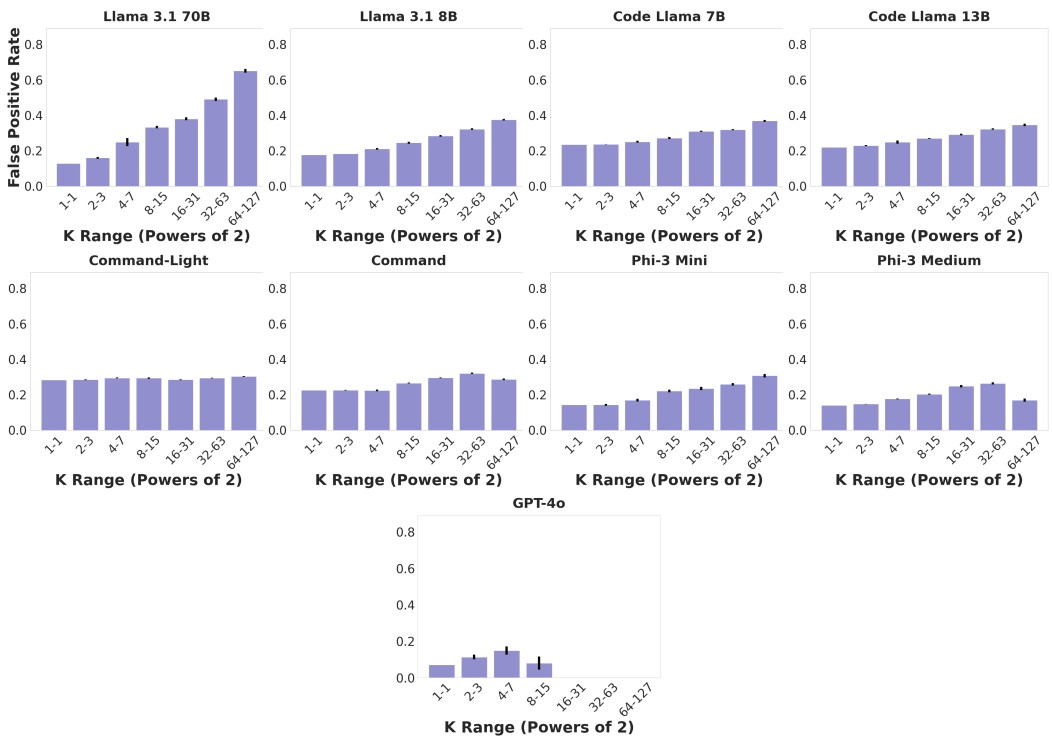

Figure 10: **False positive rates as a function of the number of attempts $K$ in HumanEval+.** $K$ ranges are aggregated into bins of increasing size (i.e., powers of 2).

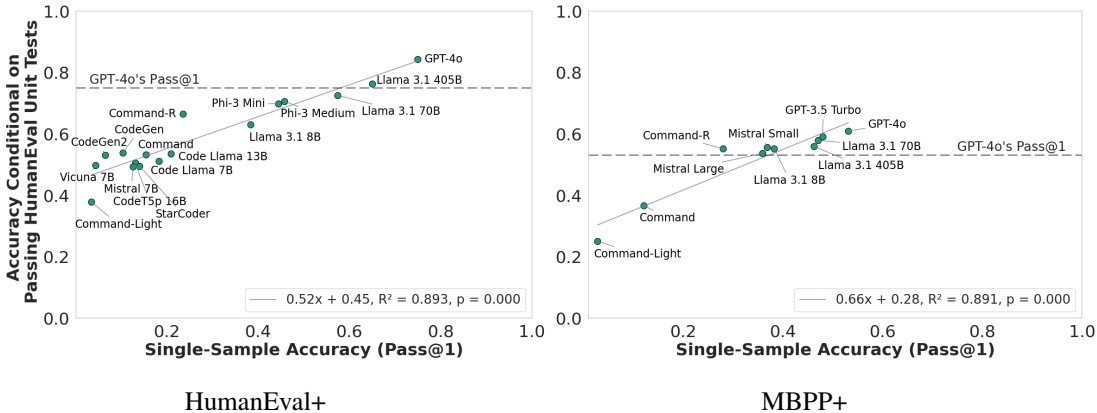

Figure 11: **Relationship between the conditional accuracy after passing the standard unit tests and single-sample accuracy for tasks on which the unit tests (i.e. the "verifier") have a precision of less than 90%.** For both benchmarks, we find a more pronounced relationship between capability and the probability of a false positive than when considering all tasks. Note that, the number of considered tasks with 70/150 and 128/321 is substantial. This plot shows the full y-axis.

## B.1 DETAILS ON DATA AND IMPLEMENTATION

In the following, we provide more details on our analysis on HumanEval+ and MBPP+.

**Sample Collection.** To evaluate the generalization gap between weaker and stronger models, we collected multiple samples per model and task. For both benchmarks and each model, we used samples generated with a temperature setting of 0.8. For sample generation, we use the implementation

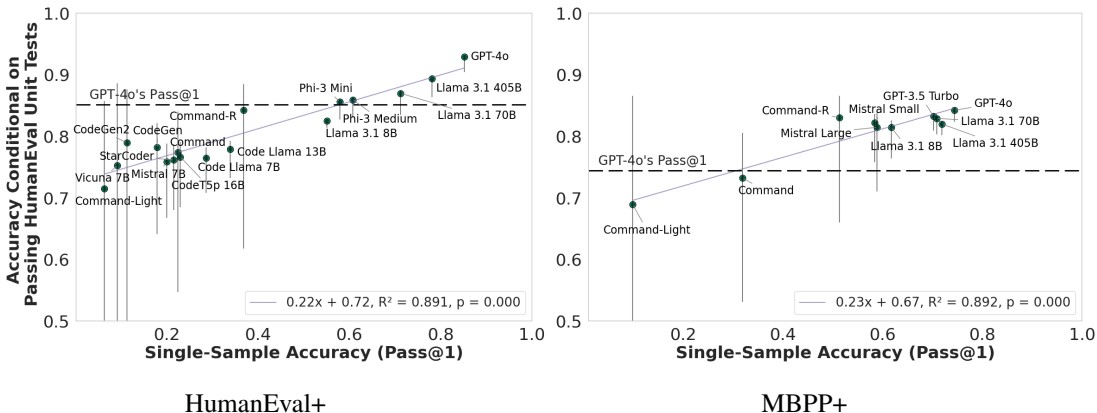

Figure 12: **Worst-case upper and lower bounds for our HumanEval+ and MBPP+ analysis.**
We show the relationship between the accuracy of individual samples (x-axis) and the achievable
accuracy given an infinite compute budget and limited unit tests (y-axis; note that it starts at 0.5).
Error bars indicate the bounds on the conditional accuracy when we account for tasks for which we
did not observe *any* passing solutions. For the upper (lower) bound, we set the conditional passing
rate to 1 (0) when computing the accuracy estimates. Note that for Command Light, even after
collecting 1000 samples for each task on HumanEval+, we still observed a substantial fraction of the
tasks without a single passing solution.

provided by Liu et al. (2023b), and other than the temperature use their default settings.[2] We collected
a minimum of 50 samples for each model and task. For most models in our experiments Vicuna
7B, Mistral 7B, CodeT5p 16B, CodeGen, CodeGen2, Code Llama 7B, and Code Llama 13B, we
used samples made available by Liu et al. (2023b). These were collected using the same temperature
setting (i.e., 0.8). We had access to 200 samples per model and task for these models. Additionally, on
HumanEval+, we collected 200 samples for Llama 3.1, Phi-3, GPT-4o, and the Command family of
models. For Command Light, we even collected 1000 samples for each task to reduce the number of
tasks without *any* solutions passing the HumanEval unit tests (Figure 12). On MBPP+, we collected
50 samples for each model and task.

### B.2 ADDITIONAL DETAILS ON EXCLUDED TASKS FROM MBPP+

**Tasks excluded by original EvalPlus authors (21 tasks).** These exclusions are based on an update
to MBPP+, during which the authors removed several broken tasks, reducing the total to 378 tasks.
These tasks were excluded because of issues with the oracle implementation leading to unreliable
evaluations [3].

In addition to the task excluded by the MBPP+ creators, in our evaluations, we excluded a total of 57
tasks from the benchmark for two main reasons:

**Tasks excluded due to additional oracle issues (28 tasks).** We identified and excluded an additional
28 tasks where *all* generated solutions that passed the base tests failed the extended test suite and
across *all* models. We used this strict criterion to ensure we would not count solutions as false
positives that are in fact robust but fail the extended test suite due to bugs in the MBPP+ harness. The
primary cause was an implementation issue in the MBPP+ oracle when handling large numerical
inputs, where the `np.allclose()` function used for checking output equivalence would raise
exceptions. After these exclusions, our final evaluation set consisted of 350 tasks from the MBPP+
benchmark.

**Tasks excluded due to solutions passing plus tests but failing standard unit tests (29 tasks).** An
additional 29 tasks were excluded for which the extended unit tests yielded passing solutions that
failed the base tests provided with the original benchmark. We intend to report these tasks to the

---

[2]See: `https://github.com/evalplus/evalplus/tree/937c46858cf8e687b31b5a728b7083d6e5a84971`
[3]See: `https://github.com/evalplus/evalplus/releases/tag/v0.3.1`

benchmark creators, aiming to include these tasks in a future version of the paper once the issue is resolved. Notably, excluding these tasks did not significantly impact our final results.

## B.3 ADDITIONAL DETAILS ON EXCLUDED TASKS FROM HUMANEVAL+

**Tasks excluded due to solutions passing plus tests but failing standard unit tests (14 tasks).** Similar to MBPP+, we excluded 14 tasks from HumanEval+ from our analysis because passing solutions on the plus tests failed the standard unit tests. Including those tasks in the analysis did, as for MBPP+, not impact our results in any significant way. We plan to report these tasks to the benchmark creators.

## C    ADDITIONAL DETAILS ON SECTION 4

In addition to the empirical analysis presented in Section 4, in this appendix, we provide a theoretical model that formalizes the limitations of inference scaling with imperfect verifiers and generalizes our findings to other benchmarks. We build on the verifier-based judge setup introduced by Davis et al. (2024). We provide a Python notebook with the implementation of our model[4].

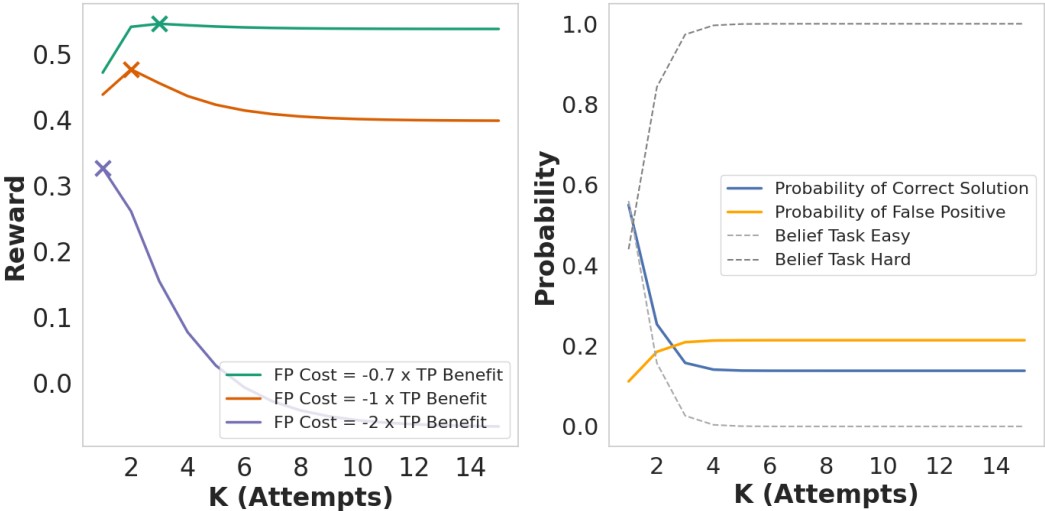

Figure 13: **Even with *zero computational cost*, the optimal number of samples is finite and very low** ($K \leq 3$). For this plot, we set the parameters as empirically observed for Llama 3.1 8B on HumanEval (see Table 3 for the exact values). The left plot shows the expected value of generating additional candidate solutions as a function of the number of attempts $K$ for various cost-benefit ratios. For all cost-benefit ratios, the expected value peaks at very low $K$, after which it begins to decline, indicating negative returns from additional sampling. The right depicts the probabilities of generating a correct solution vs. a false positive at each step $K$. As $K$ increases, the likelihood of generating a correct solution decreases, while the probability of generating a false positive increases. There is a trade-off between continued sampling and increasing risk, emphasizing the limitations of scaling inference compute with imperfect verifiers. Note that when setting the cost of a false positive to be 10 times higher than the benefit of a true positive, the optimal number of samples becomes $K = 0$ (Figure 14).

---

[4]See supplementary materials, file `limits_to_inference_scaling_model.ipynb`

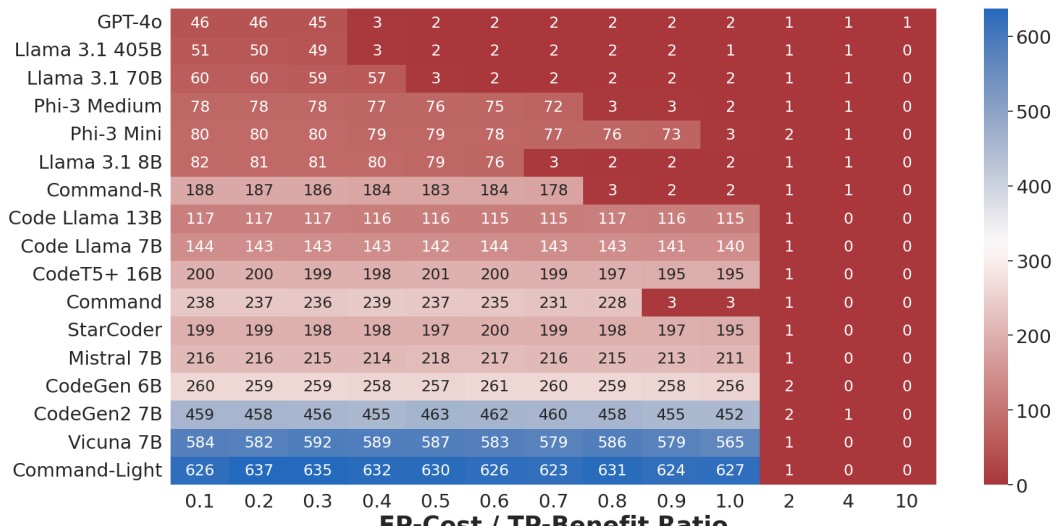

Figure 14: Heatmap of optimal number of samples $K$ for various false positive cost vs. true positive benefit ratios in our model with parameters set as observed on HumanEval+. The y-axis shows models sorted by their Pass@1 accuracy. We observe that for a relative cost of 10 (i.e., the cost of returning a false positive is 10 times more costly than the reward of returning a true positive), the optimal number of samples is $K = 0$ for almost all models, effectively making them useless.

## C.1 MODEL SETUP

The underlying model consists of two components:

- **Generator:** Produces candidate solutions to a task, with different success probabilities based on task difficulty.
  - Tasks are either *easy* ($T_1$) or *hard* ($T_2$), with prior probabilities $p_1$ and $p_2$ respectively.
  - The probabilities of generating a correct solution are $r_1$ for easy tasks and $r_2$ for hard tasks, so $r_1 > r_2$.
- **Verifier:** An imperfect verifier checks the correctness of generated solutions.
  - *Completeness* ($c$): Conditional probability of accepting a correct solution.
  - *Soundness* ($s$): Conditional probability of rejecting an incorrect solution.

PARAMETER VALUES USED IN MODEL UNDERLYING FIGURE 13

PROBABILITY OF REJECTION

The probability that a sample is being rejected by the verifier, denoted $\beta_i$, is given by:

$$\beta_i = (1 - c)r_i + s(1 - r_i) \tag{1}$$

where $i = 1$ for easy tasks and $i = 2$ for hard tasks. These probabilities ($\beta_1$ and $\beta_2$) determine how likely a generated solution is to be rejected depending on the task type.

BELIEF UPDATES

After each rejection, the belief that the task is of type $T_2$ (hard) increases. The posterior probability that the task is of type $T_1$ or $T_2$ after $k - 1$ rejections is:

$$p_{T_i}^{(k)} = \frac{\beta_i^{k-1} p_i}{\beta_1^{k-1} \cdot p_1 + \beta_2^{k-1} \cdot p_2} \tag{2}$$

| Parameter | Value |
|---|---|
| Probability of correct solution (easy task), $r_1$ | 0.87 |
| Probability of correct solution (hard task), $r_2$ | 0.13 |
| Completeness, $c$ | 1 |
| Soundness, $s$ | 0.75 |
| Prior probability of easy task, $p_1$ | 0.58 |
| Prior probability of hard task, $p_2$ | 0.42 |
| Benefit for correct solution (true positive), $V_{\text{TP}}$ | 1 |
| Cost for false positive, $V_{\text{FP}}$ | [-0.7, -1, -2] |
| Computational cost per attempt, $C_k$ | 0 |

Table 3: Parameter values used in the model setup in Figure 13 as observed for the Llama 3.1 8B model evaluated on HumanEval+. These values reflect the empirically observed probabilities and prior settings. Following the observed empirical task difficulty distribution as shown in Figure 6, in this setup we assume tasks with Pass@1 $\geq 0.5$ to be easy, and those with Pass@1 $< 0.5$ to be hard.

As more rejections occur, it usually becomes more likely that the task is hard ($T_2$). In Figure 13, we see how the belief that the task is easy decreases, while the belief that the task is hard increases as the number of attempts $K$ grows.

PROBABILITY OF CORRECT AND FALSE POSITIVE SOLUTIONS

For the $k$-th attempt, the probability of generating a correct solution or a false positive depends on the task type. The overall probabilities are weighted by the posterior beliefs $p_{T_i}^{(k)}$.

The belief-weighted probability of returning a correct or false positive at attempt $k$, conditional on the $k-1$ previous attempts being rejected are:

$$P_{\text{TP}}^{(k)} = p_{T_1}^{(k)} \cdot P_{\text{TP},T_1} + p_{T_2}^{(k)} \cdot P_{\text{TP},T_2} \tag{3}$$

$$P_{\text{FP}}^{(k)} = p_{T_1}^{(k)} \cdot P_{\text{FP},T_1} + p_{T_2}^{(k)} \cdot P_{\text{FP},T_2} \tag{4}$$

where:

$$P_{\text{TP},T_1} = c \cdot r_1, \quad P_{\text{TP},T_2} = c \cdot r_2$$

$$P_{\text{FP},T_1} = (1 - r_1) \cdot (1 - s), \quad P_{\text{FP},T_2} = (1 - r_2) \cdot (1 - s)$$

In Figure 13, the right plot shows the evolution of $P_{\text{TP}}^{(k)}$ and $P_{\text{FP}}^{(k)}$ as the number of attempts $K$ increases. Initially, the probability of generating a correct solution is higher, but for higher $K$, the probability of generating a false positive increases.

EXPECTED VALUE OF GENERATING ADDITIONAL SOLUTIONS

The expected value of generating a solution at the $k$-th attempt is:

$$\text{EV}_k = \left[ V_{\text{TP}} \cdot P_{\text{TP}}^{(k)} + V_{\text{FP}} \cdot P_{\text{FP}}^{(k)} \right] \cdot \left[ \beta_1^{k-1} \cdot p_{T_1}^{(k)} + \beta_2^{k-1} \cdot p_{T_2}^{(k)} \right] \tag{5}$$

where:

- $V_{\text{TP}}$ is the benefit for a correct solution.

- $V_{\text{FP}}$ is the cost for a false positive being "accepted" as the solution.

OPTIMAL NUMBER OF ATTEMPTS

The total expected value after $K$ attempts is:

$$\text{Reward} = \sum_{k=1}^{K} \text{EV}_k \tag{6}$$

The optimal number of attempts, $K_{\text{opt}}$, is the value of $K$ that maximizes the reward, which are shown across models and for various $V_{FP}/V_{TP}$-ratios in Figure 14.

## C.2 INFERENCE SCALING CURVE FOR GPT-4O

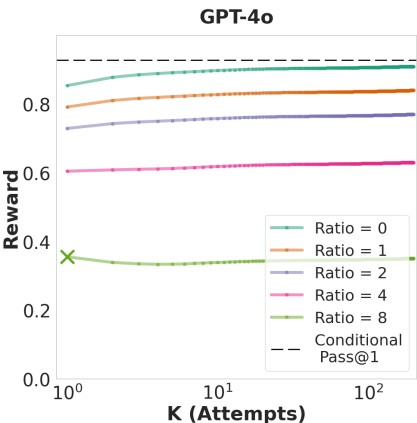

Figure 15: **Inference scaling curves in the presence of a cost for GPT-4o.** In addition to the models in Figure 1, we provide the inference scaling curves for GPT-4o as the model with the highest single-sample accuracy on on HumanEval+ in our experiments. We find that the benefits of search are minimal (i.e. curves are flat) in line with what we expect from the empirical task difficulty distribution shown in Figure 9.

# D  ADDITIONAL DETAILS ON SECTION 5

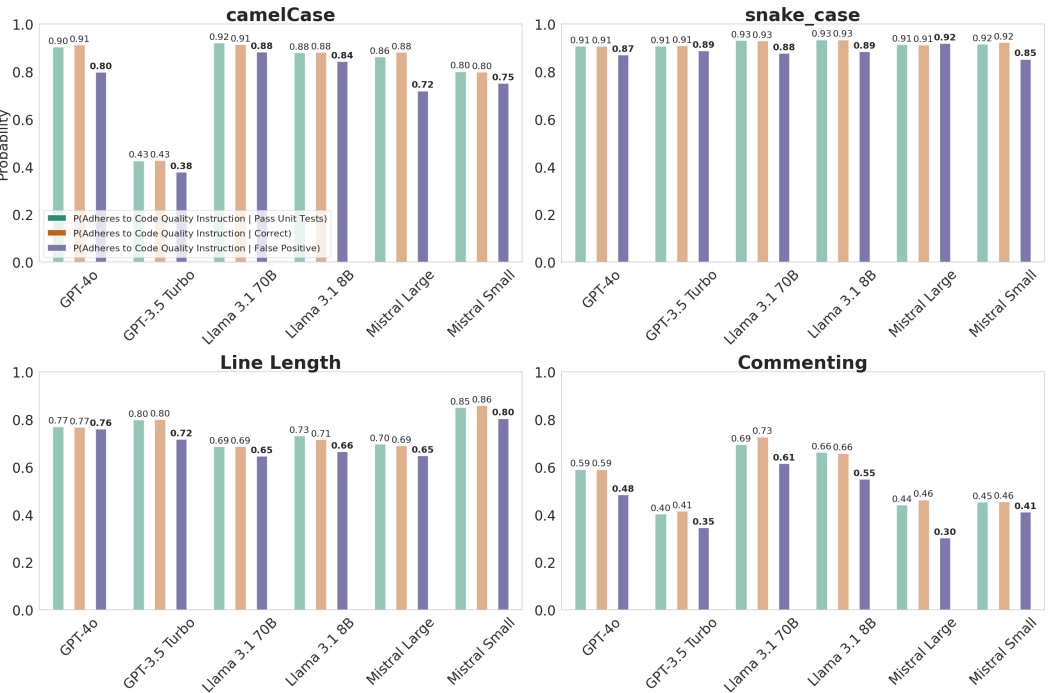

Figure 16: **False positives tend to be lower-quality code compared to correct solutions across all models and code quality metrics.** We evaluated four key code quality metrics: adherence to `camelCase` and `snake_case` naming conventions, line-length compliance, and presence of line-level comments. This trend holds consistently across models and for all four code quality instructions we test.

## D.1  DETAILS ON DATA AND IMPLEMENTATION

We used the implementation provided by Zheng et al. (2024) to collect samples and evaluate the different code readability metrics. Each code quality metric had a separate prompt instructing the model to follow certain guidelines (Section D.2). For each model and code quality instruction, we generated 50 samples per task on HumanEval+. As for our main experiments, we set the temperature to 0.8. All other parameters were set to their default value as provided with the implementation.[5]

---

[5]See: `https://github.com/jszheng21/RACE/tree/3b8ee591abd5febd8ae8ec17c7b9907949c5e1d5`

## D.2 Prompt examples for readability metrics

---

**1) Naming conventions**

Please generate the Python code to solve the following problem, and use `camelCase` for both function names and variable names.\n\nProblem:\n\n{problem}

Please generate the Python code to solve the following problem, and use `snake_case` for both function names and variable names.\n\nProblem:\n\n{problem}

**2) Code length**

Please generate the Python code to solve the following problem, where each line is less than 70 characters long and each function is less than 30 lines long.\n\nProblem:\n\n{problem}

**3) Commenting guidelines**

Please generate the Python code to solve the following problem, and add comments for each line in each function.\n\nProblem:\n\n{problem}

---

Figure 17: Prompt templates for each metric of code readability we consider in our experiments for Section 5 following Zheng et al. (2024).

## D.3 Qualitative examples of false positives

```python
def get_positive(l: list):
    return [int(item) for item in l if item > 0]
```

a) Example implementation generated by CodeGen-6B which fails additional unit tests from HumanEval+.

```python
# Standard unit tests
assert get_positive([5, 3, -5, 2, -3, 3, 9, 0, 123, 1, -10]) == [5, 3, 2,
    3, 9, 123, 1]
assert get_positive([-1, 2, -4, 5, 6]) == [2, 5, 6]

# Example unit test from extended test suite
assert get_positive([0.5, 0, -4, 2.5, 5, -2.2, -8, 7.7, 9.9, -10.5]) ==
    [0.5, 2.5, 5, 7.7, 9.9]
```

b) Standard unit tests for the HumanEval/30 task and one example test from the extended test suite of HumanEval+.

Figure 18: Example implementation that passes the standard tests but is incorrect. In HumanEval/30, the model is asked to *Return only positive numbers in the list.* The code was generated by CodeGen-6B (Nijkamp et al., 2023) and passes the tests provided with the original HumanEval benchmark but fails the extended tests. The implementation contains a logical error due to unnecessarily converting positive floats to integers.

```
1  def string_sequence(n: int) -> str:
2      ans = ""
3      for i in range(0, n + 1):
4          ans = ans + str(i) + " "
5      return ans[:-1]
```

Figure 19: HumanEval/15 asks to generate a space-delimited sequence of numbers. The implementation generated by CodeGen fails on the extended test suite due to the inefficient for-loop that times out on larger inputs.

```
1  def sort_numbers(numbers: str) -> str:
2      str_num_dict = {
3          "zero": 0, "one": 1, "two": 2, "three": 3, "four": 4,
4          "five": 5, "six": 6, "seven": 7, "eight": 8, "nine": 9,
5      }
6      num_list: List[int] = [str_num_dict[word] for word in numbers.
           split()]
7      num_list.sort()
8      sorted_str_nums: List[str] = [
9          word for word, num in str_num_dict.items() if num in
               num_list
10     ]
11     return " ".join(sorted_str_nums)
```

Figure 20: HumanEval/19 asks to sort space-delimited number words. The implementation generated by Code Llama 7B fails to maintain duplicates in the output, incorrectly removing repeated numbers during the dictionary lookup process.

```
1  def find_char_long(string
       ):
2    words = string.split()
3    return [word for word
         in words if len(word
         ) >= 4]
```

Example implementation generated by GPT-3.5 which fails additional unit tests from MBPP+.

```
1  # Standard unit tests
2  assert set(find_char_long('Please move back
       to stream')) == set(['Please', 'move', '
       back', 'stream'])
3  assert set(find_char_long('Jing Eco and Tech
       ')) == set(['Jing', 'Tech'])
4  assert set(find_char_long('Jhingai wulu road
        Zone 3')) == set(['Jhingai', 'wulu', '
       road', 'Zone'])
5
6  # Example unit tests from extended test
       suite
7  assert set(find_char_long("This is a very
       long string with no words that are at
       least 4 characters long.")) == set(['
       This', 'very', 'long', 'string', 'with',
        'words', 'that', 'least', 'characters',
        'long'])
```

Standard unit tests for the MBPP/7 task.

Figure 21: Exemplary implementation that passes the standard tests but is incorrect. In MBPP/7, the model is asked to *write a function to find all words that are at least 4 characters long in a string.* The code was generated by gpt-3.5 and passes the tests provided with the original MBPP benchmark but is incorrect and fails to pass the extended tests.

```python
def parse_nested_parens(paren_string: str) -> List[int]:
    def parse_paren_group(paren_group: str) -> int:
        counter = 0
        max_counter = 0
        for paren in paren_group:
            if paren == '(':
                counter += 1
                if counter > max_counter:
                    max_counter = counter
            elif paren == ')':
                counter -= 1
        return max_counter
    return [parse_paren_group(paren_group) for paren_group in
        paren_string.split(' ')]
```

Figure 22: HumanEval/6 requires parsing nested parentheses groups. The implementation generated by Code Llama 7B incorrectly processes empty input strings.

```python
def sorted_list_sum(lst):
    assert lst, "List can not be empty."
    if isinstance(lst[0], str):
        new_lst = [word for word in lst if len(word) % 2 == 0]
        new_lst.sort()
        return sorted(new_lst, key=len)
    else:
        return sorted(lst)
```

Figure 23: HumanEval/149 asks to process a list of strings, removing odd-length strings and sorting the result. The implementation generated by CodeT5p 16B incorrectly handles empty input by raising an assertion error instead of returning an empty list.

```python
def countNumbersWithOne(n):
    if n == 1:
        # For single-digit numbers, we only have 1
        return 1

    count = 0

    # Calculate the range for n-digit numbers
    start = 10**(n-1)
    end = 10**n

    for num in range(start, end):
        str_num = str(num)
        if str_num[0] == '1' or str_num[-1] == '1':
            count += 1

    return count
```

Figure 24: Example candidate implementation failing to adhere to `camelCase` naming convention for HumanEval/83 generated by GPT-4o due to variable *str_num* in line 13 not complying.

```
1  def get_positive(l: list):
2      return [int(item) for item in l if item > 0]
```

Figure 25: HumanEval/30 asks to return positive numbers from a list. The implementation generated by CodeGen fails on the extended test suite because it converts float values to integers before returning the list.

```
1  def number_of_substrings(s):
2      char_count = [0] * 128
3      result = 0
4      for char in s:
5          char_count[ord(char)] += 1
6      for i in range(len(s)):
7          result += char_count[ord(s[i])] * (len(s) - i)
8      return result
```

Figure 26: MBPP/238 asks to count non-empty substrings in a string. The implementation generated by Command uses an incorrect algorithm that produces wrong results.

```
1  def split(string):
2      return [char for char in string if char.isalpha()]
```

Figure 27: MBPP/565 asks to split a string into characters. The implementation generated by Command Light incorrectly filters for alphabetic characters only and fails to handle empty strings correctly.

```
1  def is_samepatterns(words, patterns):
2      if len(words) != len(patterns):
3          return False
4      word_to_pattern = {}
5      pattern_to_word = {}
6      for word, pattern in zip(words, patterns):
7          if word in word_to_pattern and word_to_pattern[word] !=
               pattern:
8              return False
9          if pattern in pattern_to_word and pattern_to_word[pattern]
                != word:
10             return False
11         word_to_pattern[word] = pattern
12         pattern_to_word[pattern] = word
13     return True
```

Figure 28: MBPP/74 asks to check if words follow given patterns. The implementation generated by GPT-3.5 fails on the extended test suite when handling repeated elements, incorrectly returning False.

```
1  def find_even_pair(nums):
2      even_count = 0
3      xor_pairs = set()
4      for i in range(len(nums)):
5          for j in range(i+1, len(nums)):
6              if (nums[i] ^ nums[j]) % 2 == 0:
7                  pair = tuple(sorted((nums[i], nums[j])))
8                  xor_pairs.add(pair)
9      return len(xor_pairs)
```

Figure 29: MBPP/166 asks to count pairs of integers with even XOR. The implementation generated by Llama 3.1 8B incorrectly counts only unique pairs using a set.

```
1  def diff_even_odd(numbers):
2      even = None
3      odd = None
4      for num in numbers:
5          if num % 2 == 0 and even is None:
6              even = num
7          elif num % 2 == 1 and odd is None:
8              odd = num
9          if even is not None and odd is not None:
10             break
11     return abs(even - odd)
```

Figure 30: MBPP/594 asks to write a function to find the difference between the first even and the first odd number of a given list. The implementation generated by GPT-3.5 incorrectly returns the *absolute* value of the difference.

## E    DECLARATION OF LLM USAGE AND COMPUTE RESOURCES

In this work, LLMs are the main subject of study and used for sampling solutions for coding benchmarks. For all our experiments using OpenAI models, we utilized the endpoints provided by OpenAI, either directly or through the Azure OpenAI Service. For the analysis on HotPotQA using Llama models, we relied on the endpoints provided by Together.ai. As our work primarily relied on external APIs, we did not use any GPUs for inference and our experiments did not require training of LLMs.

## F    IMPACT STATEMENT

Our work aims to enhance the technical understanding of the limitations of inference scaling methods. In particular, we contribute new findings on resampling in combination with imperfect verifiers and how this approach can fail to lift the accuracy of weaker models to match the performance of stronger models. While these findings have implications for the training of reasoning models and the deployment of compound AI systems—including code generation tools—their broader ethical and societal implications mirror those already familiar in the development and use of large-scale language models. We do not identify any additional, domain-specific concerns that arise uniquely from our study. Instead, our results reinforce the importance of reliable evaluation metrics and thorough verification methods, which in turn support safer and more trustworthy applications in coding as well as in other areas where compound AI systems are increasingly adopted.

