# OpenReview forum: "The Limits of Inference Scaling Through Resampling"
_ICLR.cc/2026/Conference — ICLR 2026 Poster_

### Official Review · Reviewer_Tt4Y · 2025-10-31

**Soundness:** 3
**Presentation:** 3
**Contribution:** 3
**Rating:** 6
**Confidence:** 3

**Summary:**

The paper examines inference-time “resampling with verifiers” and shows that when verifiers are imperfect (i.e., admit false positives), weaker models cannot sample their way up to a strong model’s single-sample accuracy. On HumanEval+ and MBPP+, many solutions from weaker models that pass standard unit tests fail extended tests, so even with unlimited samples, they remain below a strong model's Pass@1. Incorporating the cost of false positives further implies that the optimal sample count is small. The authors also show that false-positive solutions are lower-quality code beyond correctness (naming, comments, length).

**Strengths:**

1. The central message is clearly motivated and explained: imperfect verifiers induce an upper bound on performance gains from resampling, and weaker models cannot match stronger models’ Pass@1 via increased sampling.

2. The extensive experiment demonstrates a few interesting observations: 1) the ceiling effect persists across model families and sampling budgets, 2) the optimal sample count is typically small when false positives incur penalties, and 3) false positive solutions exhibit systematically poorer code properties (naming, comments, verbosity).

**Weaknesses:**

1. The scope of the paper is coding-centric. The arguments are persuasive for unit-test verifiers, but other imperfect verifiers, such as LM-judges for QA/reasoning, could also be studied.

2. Resampling changes the distribution of candidate solutions; the verifier’s FPR may not be stationary with K (e.g., later samples are weirder/hackier). Do you observe drift in the verifier errors as K grows?

3. A single verifier (tested in the paper) would create a brittle FPR. What is the ceiling under, for instance, majority ensembles of various verifiers? In that case, how does diversity (e.g., correlation of errors) mathematically translate into ceiling relief?

**Questions:**

1. If the verifier can abstain at a tunable threshold to bound FPR$\leq \varepsilon$, how does the ceiling shift? Can you train a “selective verifier” with coverage guarantees and show a different optimal K frontier?

2. The cost of a false positive is treated uniformly in the paper. In practice, costs vary by task and downstream impact. Is it possible to learn task-conditioned costs (e.g., via proxy severity scores) and optimize K per instance?

3. How does sampling configuration (e.g., top-p/temperature) affect the conclusions?

---

> ### Author Response · Authors · 2025-11-24
>
> We thank the reviewer for taking the energy and time to review our paper. We especially appreciate the comments on abstention and calibrating verifiers.
>
> *The scope of the paper is coding-centric.*
>
> We acknowledge that our experiments focus solely on repeated sampling in the context of coding tasks, as discussed in Section 6\. While Coding offers a clear example of the challenges posed by imperfect verifiers, other domains might exhibit different behavior.
>
> However, we also emphasize that the theoretical model presented in Appendix C is domain-agnostic. As long as a verifier is imperfect, our findings generalize directionally, although the effect size might differ. As noted in Section 6 and Table 2, imperfect verification is a pervasive issue in reasoning-heavy tasks and agentic workflows. We view the coding experiments as a clean empirical demonstration of the phenomenon in one of the key application-relevant domains.
>
> *Resampling changes the distribution of candidate solutions; the verifier’s FPR may not be stationary with K (e.g., later samples are weirder/hackier).*
>
> As discussed in Section 4 (Page 7), the distribution of task difficulty is bimodal. As k  increases, the remaining pool of unsolved tasks shifts towards "harder" tasks. Figure 5 shows that the FPR tends to increase with k precisely because of the underlying shift in task distribution.
>
> *A single verifier (tested in the paper) would create a brittle FPR. What is the ceiling under, for instance, majority ensembles of various verifiers?*
>
> We agree that ensembles could improve performance, effectively creating a "stronger" verifier. However, our core argument is about the limits imposed by imperfection (P(False Positive) \> 0). Even an ensemble is unlikely to be a perfect oracle. Therefore, the asymptotic ceiling described in Figure 1 and Section 3 remains, though it might be shifted higher.
>
> *The cost of a false positive is treated uniformly in the paper. In practice, costs vary by task and downstream impact. If the verifier can abstain at a tunable threshold to bound FPR, how does the ceiling shift?*
>
> This is a thoughtful point, and we recognize that more realistic cost calculations would include heterogeneity across tasks. In fact, abstention would be a great strategy to mitigate the incurred cost of false positives and mitigate the issue we study for practical applications. Our core argument is about the limits imposed by imperfection (P(False Positive) \> 0). Even a tunable system for abstention will unlikely become a perfect oracle, albeit achieving satisfactory practical performance.
>
> *How does sampling configuration (e.g., top-p/temperature) affect the conclusions?*
>
> As per Appendix B.1, we used a temperature of 0.8 for all experiments to maintain consistency with previous benchmarks (Liu et al., 2023b). While we did not perform a sweep over sampling parameters, we hypothesize that lower temperatures (e.g., 0.2) might slightly reduce the false positive rate by reducing rare generation, but would simultaneously reduce the coverage (Pass@K) for harder problems that require exploration. Liu et al. (2023b) did a temperature sweep on the evalplus benchmarks and found 0.8 to be the best-performing setting to improve Pass@K.
>
> We hope these points address some of the concerns of the reviewer and are happy to provide any further clarification or analysis.

---

### Official Review · Reviewer_JXaJ · 2025-11-01

**Soundness:** 3
**Presentation:** 3
**Contribution:** 3
**Rating:** 6
**Confidence:** 3

**Summary:**

This paper argues that inference-time scaling via repeated sampling + imperfect verifiers (like unit tests) has a hard ceiling. Even if you spend infinite inference compute, weaker models cannot necessarily catch up to stronger models, because false positives (incorrect-but-accepted answers) become the bottleneck. The paper backs this up with theory and experiments on HumanEval+/MBPP+, and also studies utility tradeoffs when false positives are costly, plus code quality effects.

**Strengths:**

1. The paper challenges a widely held assumption (“a weak model can just sample more and pass tests to match a stronger model”) and shows that imperfect verifiers are actually the fundamental bottleneck. This is framed cleanly and feels very relevant right now.

2. Using benchmarks like HumanEval+ and MBPP+, the paper quantifies how weaker models produce many false positives (wrong answers that still pass tests) and shows that even with unlimited resampling they hit a ceiling and still can’t match stronger models.

3. The paper doesn’t stop at benchmark accuracy — it talks about real-world cost of shipping bad code, the risk of contaminating training data with verifier-approved mistakes, and why “just sample more” is not a reliable path to replacing stronger models in production.

4. The figures and tables (e.g. showing the accuracy ceiling line, and how the optimal number of attempts K is actually low when false positives are costly) make the main ideas easy to understand and memorable

5. The experimental ideation in Section 4 is novel. Although the assumptions regarding the cost are simplified, this is inevitable. The experiment provides a practical and insightful analysis of the cost associated with infinite or large-scale sampling.

**Weaknesses:**

Overall, I appreciate the quality of the paper, though I have several points of concern outlined below.

1. Because the verifier’s effectiveness depends on the specific test suite, its accuracy and false-positive behavior may vary across different sets of test cases. The paper would benefit from an analysis of how the results change when the verifier’s quality or coverage is systematically varied.

2. A considerable amount of the introductory section (up to approximately page 4) is spent discussing scaling methods beyond the imperfect verifier. It is somewhat questionable whether this level of detail is essential, given that the paper’s primary scope focuses on tasks involving imperfect verifiers in the coding domain. Such elaborations might be better suited for the appendix.

**Questions:**

Question and suggestion are listed in weakness.

---

> ### Author Response · Authors · 2025-11-24
>
> We thank the reviewer for their positive assessment and constructive feedback.
>
> *Because the verifier’s effectiveness depends on the specific test suite, its accuracy and false-positive behavior may vary across different sets of test cases.*
>
> This is very thoughtful. We were mitigating this variable with a two-fold strategy when designing our analysis:
>
> 1. In our theoretical analysis (Appendix C), we explicitly modeled the quality of the verifier in our theoretical framework using the soundness parameter (s), defined as the conditional probability of rejecting an incorrect solution (Table 3). We provide a theoretical model that formalizes the limitations of resampling we observe empirically and generalizes our findings to other benchmarks.
> 2. Empirically, we observe consistent behavior across two different benchmarks, HumanEval and MBPP. This suggests that our findings regarding the limits of resampling are not artifacts of one specific test suite, but a general property of using imperfect verifiers with resampling.
>
> *A considerable amount of the introductory section (up to approximately page 4\) is spent discussing scaling methods beyond the imperfect verifier.*
>
> We recognize this and will update the manuscript to be more concise. We felt the need for the conceptual differentiation to make clear that we specifically look at verification in this paper and to distinguish it from reranking, majority voting, etc., which are sometimes treated interchangeably in the literature. More details on this are in Appendix A.1.
>
> We hope we could clarify and mitigate some of the reviewer’s feedback. Thanks again for taking the time to review.

---

### Official Review · Reviewer_9qjn · 2025-11-03

**Soundness:** 2
**Presentation:** 2
**Contribution:** 3
**Rating:** 4
**Confidence:** 3

**Summary:**

This paper explores the limitations of scaling resampling in two coding benchmarks and shows that weaker models are more prone to generating false positives—solutions that pass some unit tests but fail when subjected to more comprehensive testing. The study reveals that, in the coding domain, the effectiveness of resampling is fundamentally constrained by the insufficiency of unit tests, a challenge frequently encountered in real-world applications

**Strengths:**

The main finding is compelling: Limited test cases constrain the scalability of resampling, and weaker models experience a larger generalization gap than stronger ones. Notably, most weak models cannot outperform the single-sample accuracy of stronger models even when resampling is scaled up. These results highlight the inherent limitations of resampling in real-world coding scenarios, where test cases are typically limited.

The analyses are insightful: By introducing the concept of the cost of false positives and evaluating the quality of generated code, the authors illuminate the potential risks associated with applying resampling techniques in practical coding applications.

The experiments are comprehensive, spanning various model families and two benchmarks.

**Weaknesses:**

- The use of a large table (Table 1) to list numerous inference scaling techniques feels excessive, particularly since this paper only analyzes the limitations of one specific technique. Including such an extensive table for all techniques seems unnecessary and distracts from the actual focus of the study. A similar concern applies to Table 2.

- The authors attempt to extend their findings to broader domains; however, the paper’s analysis is limited to the coding domain. While test cases serve as verifiers in coding, other domains (e.g., mathematics) might utilize reward models, which differ fundamentally. Although both are imperfect verifiers, the distinctions between test cases and reward models are substantial, making broad generalization of the findings from
coding to other domains unconvincing

Insufficient justification for main finding:

- (a) The central claim that “the weaker model cannot match the performance of a single invocation of the stronger model” is presented too absolutely. For instance, Figure 3 shows cases where Llama 3.1 70B outperforms a single invocation of GPT-4o, and Llama 3.1 8B outperforms a single invocation of Llama 3.1 70B, contradicting the claim.
- (b) In line 246, the authors state they generate at least 50 samples for each model and benchmark. The rationale for the “at least” threshold is unclear. Why is it not standardized? How are the actual sampling counts determined?
- (c) In lines 261–263, the statement that the finding holds “no matter how big the compute budget for the weaker model” is insufficiently supported. Only experiments with “at least 50 samples” are reported, leaving the maximum compute budget used ambiguous and raising the question of whether a larger compute budget could yield different results.

**Questions:**

See the weakness

---

> ### Author Response · Authors · 2025-11-24
>
> We want to thank the reviewer for their thoughtful comments and feedback and provide some responses and actions in the following.
>
> *The use of a large table (Table 1\) to list numerous inference scaling techniques feels excessive*
>
> * Table 1 is necessary to position Resampling with Verifiers (our focus) within the broader landscape of inference scaling (e.g., distinct from Majority Voting or Chain-of-Thought), clarifying exactly which inference scaling strategy faces the limitations we identify
> * Table 2 emphasizes the fact that "Imperfect Verification" is not a niche issue but a widespread standard in the literature. It demonstrates that reliance on imperfect proxies (like LM-as-a-judge or partial unit tests) is the norm across recent work. This validates our findings.
>
> Based on this feedback, we update the paper to stick with the most important conceptual points and inference scaling techniques in the tables to stay more concise overall.
>
> *While test cases serve as verifiers in coding, other domains (e.g., mathematics) might utilize reward models, which differ fundamentally.*
>
> Thanks for this comment. We do agree with that and emphasize that we look at verification specifically in this paper. We try to make this differentiation between reranking and verification in Figure 2\. So while RM can be considered imperfect verifiers when paired with an acceptance threshold, we differentiate conceptually to provide clarity on all the nuances of inference scaling techniques.
>
> We will make sure to update the manuscript to be clearer.
>
> *Insufficient justification for main finding*
>
> a) We believe there is a misunderstanding regarding the interpretation of Figure 3\. The reviewer states: "Llama 3.1 70B outperforms a single invocation of GPT-4o... contradicting the claim.”
>
> Correction: In Figure 3 (Left, HumanEval+), the horizontal dashed line represents GPT-4o's Pass@1 (approx. 0.62 on the x-axis). The Llama 3.1 70B data point (orange 'X') has a y-value (conditional accuracy) that is below this dashed line. This supports our claim: even with "infinite" resampling (the y-value limit), Llama 3.1 70B cannot match the single-shot performance of GPT-4o.
>
> b) This was a phrasing issue in the main text. As detailed in Appendix A2.1 and A5.1, we generated 200 samples per model and task on HumanEval+ and 50 samples per model and task on MBPP+. Only for Command-Light, we generated 1,000 samples in order to minimize the number of tasks without any passing solutions. We will update the paper to phrase this more accurately in the main text.
> c) Our "infinite compute" metric is a measurement of precision. We define the limit of inference scaling as P(Correct∣Pass Verifier). This probability represents the performance ceiling: once a model generates a solution that passes the verifier, resampling stops. If the verifier accepts incorrect code (FP), no amount of additional compute can correct that error. By generating 200+ samples, we obtained a robust estimate of this conditional probability (the verifier's precision) because we sampled passing solutions for most models on almost all tasks of the benchmarks. We also offer worst-case error bars in the Appendix on what happens if all passing samples for outstanding tasks would end up being True Positives for each model (very unlikely) and show that our findings still hold.
>
> We hope these clarifications resolve the reviewer's concerns, particularly regarding the interpretation of Figure 3 and the sample sizes used to validate our empirical bounds.

---

### Official Review · Reviewer_Tash · 2025-11-06

**Soundness:** 3
**Presentation:** 3
**Contribution:** 3
**Rating:** 6
**Confidence:** 5

**Summary:**

This paper presents a critical analysis of inference-time scaling, specifically the "resampling" approach where solutions are generated until one passes a verifier. The authors argue that this technique is **fundamentally limited** when the verifier is imperfect (i.e., has a non-zero false positive rate).

The core contributions are:
1.  An empirical demonstration on coding benchmarks (HumanEval+ and MBPP+) that a model's single-sample accuracy (Pass@1) is **strongly correlated with its false positive rate**. Weaker models produce *more* false positives (solutions that pass limited unit tests but fail comprehensive ones) than stronger models.
2.  This correlation establishes a **hard upper bound** on the performance of resampling. A weaker model, even with an infinite compute budget, cannot match the single-sample accuracy of a stronger model if the stronger model's accuracy is already higher than the weaker model's *conditional* accuracy (its accuracy on solutions that pass the imperfect verifier).
3.  By introducing a "cost" for false positives, the paper shows that the optimal number of sampling attempts (K) is often **finite and very low** (e.g., fewer than 10), as the expected reward curve bends downward when the risk of a false positive outweighs the benefit of finding a true positive.
4.  A secondary finding that these false positive solutions are not just functionally incorrect but are also of **lower code quality** (e.g., poor adherence to style conventions).

**Strengths:**

* **Novel, Important, and Non-Obvious Finding:** The paper's main strength is the empirical discovery of the strong linear correlation between a model's Pass@1 accuracy and its false positive rate. This is not an obvious or trivial result; one might have assumed the "imperfect verifier" was an independent challenge. This finding is the engine for all the paper's other conclusions.
* **Elegant and Sound Methodology:** As mentioned in "Soundness," the use of the `Benchmark` vs. `Benchmark+` test suites is a perfect experimental setup to test the hypothesis.
* **Strong, Clear-Cut Results:** The data presented is not ambiguous. The linear relationship is strong ($R^2 \approx 0.89$), and the performance ceiling is clearly visible. The downward-bending reward curves are also unambiguous.
* **Practicality of the Critique:** The paper addresses a real-world, practical strategy (resampling) and exposes its fundamental limitations in a way that is immediately useful for both researchers and practitioners.
* **Secondary Quality Analysis:** The analysis in Section 5, showing that false positives are *also* lower-quality code (e.g., worse style), is a strong supporting argument that adds another dimension to the critique.

**Weaknesses:**

* **Static, Domain-Specific Verifier:** The paper's *entire* analysis hinges on a **static, human-written verifier** (the original HumanEval/MBPP unit tests) in the single domain of **coding**. The central claim that resampling is "fundamentally limited" is very strong, but the evidence is scoped to this specific setup. The paper's claim of being "domain-agnostic" is not well-supported, especially since Table 2 explicitly lists "Math" as a domain where "Oracle" verifiers (like proof checkers) *do* exist, which would make resampling a perfectly valid strategy in that domain.
* **Doesn't Account for Scaling Verifiers:** The analysis assumes the *generator* scales (from Llama 7B to GPT-4o) but the *verifier* is fixed. In many modern systems, the verifier is *also* an LLM (e.g., "LM-as-judge") or is *generated* by the model itself (e.g., model-generated unit tests). A stronger model is likely also a stronger verifier. The paper's conclusions might not hold in a "co-scaling" scenario where the verifier's capability (and thus its false positive rate) improves along with the generator's. The paper mentions this but does not investigate it, which is the most significant limitation of this work.
* **Ambiguity of "Cost":** The analysis of the optimal K in Section 4 is highly dependent on the "cost-benefit ratio," a variable for which the paper provides no empirical grounding. While it's intuitive that a bug (false positive) has a high cost, showing a range from 0 to 8 makes the finding abstract. Without a "realistic" C/B ratio, it's difficult to conclude whether the optimal K for a real-world application is 3 or 300.
* **Significant Task Exclusion:** The appendix reveals that a large number of tasks were excluded from the analysis (e.g., 78 from MBPP+ and 14 from HumanEval+) due to issues with the test harness. The paper asserts this "did not significantly impact our final results", but this requires the reviewer to trust this claim. It's plausible that these "broken" or ambiguous tasks are precisely the ones that would most effectively probe the limitations of weaker models.

**Questions:**

1.  **Scaling Verifiers:** Your analysis relies on a static, human-written verifier. How do you believe your findings would change in a setting where the verifier also scales? For example, what if you used GPT-4o as an "LM-as-judge" to verify the solutions from all models, or used each model to generate its *own* unit tests?
2.  **Domain-Agnostic Claim:** You claim the results are "domain-agnostic", yet your own Table 2 shows that "oracle" verifiers are common in mathematics. Can you clarify your claim? Does the "limit" you've identified primarily apply to domains *lacking* formal, oracle verifiers (like coding with incomplete tests, or creative writing)?
3.  **Cost-Benefit Ratio:** Your "optimal K" analysis is very sensitive to the C/B ratio. Do you have any suggestions for how one might empirically estimate a "realistic" C/B ratio for a typical software engineering task, to move this finding from a theoretical observation to a practical prescription?
4.  **Task Exclusion:** You excluded 78 tasks from MBPP+ due to test harness issues. Can you provide more detail on why you are confident this did not bias the results? For instance, did you check if the Pass@1 or conditional accuracy *on this subset of tasks* showed a different trend?

---

> ### Author Response · Authors · 2025-11-24
>
> We thank the reviewer for their feedback and for taking the time to review the paper.
>
> *The paper's conclusions might not hold in a "co-scaling" scenario where the verifier's capability improves along with the generator's.*
>
> Thanks for this comment and observation. We explicitly acknowledge Math as an exception with oracle verifiers (Table 2); our findings apply to the vast majority of domains that lack them. More precise verifiers are certainly a valuable direction, but they do not overcome the underlying limitations shown in our findings. Unless the scaled verifier achieves FPR=0, the ceiling remains, albeit shifting upward.
>
> *Does the "limit" you've identified primarily apply to domains lacking formal, oracle verifiers (like coding with incomplete tests, or creative writing)?*
>
> Thanks for this clarifying question. Yes, In Section 2 and specifically Table 1, we are drawing the conceptual line between imperfect / oracle verification as well as verification and other inference scaling strategies such as reranking or majority voting. Our paper specifically studies imperfect verification. Many domains such as coding, math, QA, etc. (Table 2\) are using imperfect verification for inference scaling which underscores its real-world relevance.
>
> *The analysis of the optimal K is highly dependent on the "cost-benefit ratio," a variable for which the paper provides no empirical grounding... difficult to conclude whether the optimal K for a real-world application is 3 or 300\.*
>
> We thank the reviewer for this valid observation. We agree that determining a precise, universal Cost-Benefit ratio is difficult, as it varies significantly by application. However, we believe our findings remain robust and actionable for the following reasons:
>
> * In realistic coding tasks, the cost of a FP is almost surely \> 0\. In our theoretical model in Appendix C, as well as empirically (Section 4 or Figure 1), we show that even for very modest CB ratios, the optimal sampling limit drops precipitously (e.g., from limiting out at 200 down to single digits).
> * The prevailing assumption in inference scaling is that one can simply sample indefinitely until a passing solution is found (Brown et al., 2024). Our work demonstrates that this promise falls through when using imperfect verifiers (like unit tests); for which the optimal number of samples is not only finite but tends to be very low.
>
> We will update the manuscript to explicitly state that the specific K values depend on the practical context and note it as a potential limitation.
>
> *You excluded 78 tasks from MBPP+ due to test harness issues. Can you provide more detail on why you are confident this did not bias the results?*
>
> These exclusions prevent bias rather than introduce it. We removed tasks based on objective errors in the benchmark harness, not model performance. For example, in 28 cases, the benchmark's own *oracle solution* failed the extended tests. Including these would penalize models for correctness. In order to validate our findings against worst-case scenarios where questions for which we didn’t sample any correct generations would all be TP, we provide worst-case error bars in Figure 12\.
>
> We hope these points resolve some of the reviewer’s concerns. We thank the reviewer again for their valuable feedback!

---

### Meta-Review · Area_Chair_1jYf · 2026-01-06

**Summary:**

All the reviewers praised the novel and practical finding that imperfect verifiers create a hard ceiling for resampling. The key insight is that weaker models cannot match stronger models' Pass@1 due to false positive bottlenecks. The methodology was generally perceived as elegant and the multifaceted analysis and experiments were praised by the reviewers.

The issues lied with overgeneralization of the domain (making domain-agnostic claims were considered problematic). Some of the examples served as counterexamples for the claims and some gaps in methodology were observed.

Given the two different views, the paper is truly on the borderline.

**Reviewer Concerns:**

Some issues about clarity have been addressed in teh rebuttal. Some questions about overgeneralization have been acknowledged.

**Reviewer Scores:**

It is unclear to me if the authors would have fully convinced reviewer 9jqn. At most, I can see a bump to 5. I cannot see anyone championing the paper but generally the paper is on the borderline.

---

### Decision · Program_Chairs · 2026-01-26

Accept (Poster)